# Landscape evolution models using the stream power incision model show unrealistic behavior when *m/n* equals 0.5

Jeffrey S. Kwang[1], Gary Parker[1,2]

[1]Department of Civil and Environmental Engineering, University of Illinois at Urbana-Champaign, Urbana, IL, USA
[2]Department of Geology, University of Illinois at Urbana-Champaign, Urbana, IL, USA

*Correspondence to*: Jeffrey S. Kwang (jeffskwang@gmail.com)

**Abstract.** Landscape evolution models often utilize the stream power incision model to simulate river incision: $E=KA^mS^n$, where $E$ = vertical incision rate, $K$ = erodibility constant, $A$ = upstream drainage area, $S$ = channel gradient, and $m$ and $n$ are exponents. This simple but useful law has been employed with an imposed rock uplift rate to gain insight into steady-state
landscapes. The most common choice of exponents satisfies $m/n = 0.5$. Yet all models have limitations. Here, we show that when hillslope diffusion (which operates only at small scales) is neglected, the choice $m/n = 0.5$ yields a curiously unrealistic result: the predicted landscape is invariant to horizontal stretching. That is, the steady-state landscape for a 10 km$^2$ horizontal domain can be stretched so that it is identical to the corresponding landscape for a 1000 km$^2$ domain.

## 1 Introduction

The stream power incision model (SPIM) (e.g., Howard, 1994; Howard et al., 1994) is a commonly-used physically-based model for bedrock incision. The incision rate, $E$, can be written as

$$E = KA^mS^n \tag{1}$$

where $K$ = erodibility coefficient, $A$ = upslope drainage area, $S$ = downstream slope, and $m$ and $n$ are exponents. This simple model is thoroughly reviewed in Whipple and Tucker (1999) and Lague (2014), where they hypothesize that $m/n$ is between
0.35 and 0.60. This range is consistent with results inferred from field work and map studies (Flint, 1974; Howard and Kerby, 1983; Tarboton et al., 1989; Willgoose et al., 1990; Tarboton et al., 1991; Willgoose, 1994; Moglen and Bras, 1995; Snyder et al., 2000). Furthermore, many researchers specifically suggest, or offer as a default, the ratio, $m/n \sim 0.5$ (Snyder et al., 2000; Banavar et al., 2001; Hobley et al. 2017). The choice of this ratio is paramount in numerical Landscape Evolution Models (LEMs) that utilize SPIM, such as the channel-hillslope integrated landscape development model, *CHILD* (Tucker et al., 2001).
The ratio, $m/n$, is also used to describe the relationship between slope and drainage area in describing stream long profiles (Flint, 1974). All models using SPIM, including studies on drainage reorganization and stability (Willett et al., 2014), tectonic histories of landscapes (Goren et al., 2014b; Fox et al., 2014), and persistent drainage migration (Pelletier, 2004), involve specification of this ratio. In addition, the specific values of $m$ and $n$ are important (Tucker and Whipple, 2002). Here, however, we focus on the ratio itself, and we show a somewhat unexpected result: when $m/n = 0.5$, SPIM-based LEMs exhibit elevation

solutions that are invariant to shape-preserving stretching of horizontal domain. That is, except for the finest scales at which hillslope diffusion becomes important, the model predicts the same solution for a landscape with a total basin area of 10 km$^2$ and one with a total basin area of 1000 km$^2$ under the constraint of identical horizontal basin shape (e.g. square). The extremity of this result underscores a heretofore unrecognized unrealistic aspect of SPIM.

In this paper, we perform a scaling analysis of SPIM. First, we use a 1D model to analytically derive steady-state river profiles, to illustrate the problem of scale invariance, and to delineate conditions for which elevation singularities occur at the ridge. Then, using a 2D numerical model, we demonstrate the effects of horizontal scale on the steady-state relief of landscapes and infer the conditions for which elevation singularities occur at ridges.

**2 Motivation**

SPIM is a simple model that has been used to gain considerable insight into landscape evolution. Previous studies using SPIM have shown how landscapes respond to tectonic and climate forcing (e.g., Howard, 1994; Howard et al., 1994). Yet like most simple models, SPIM is in some sense an oversimplification. Here we demonstrate this by showing that it satisfies a curiously unrealistic scale invariance relation. By demonstrating this limitation, we hope to motivate the formulation of models that
overcomes it.

The fundamental limitation on SPIM becomes apparent when the ratio, $m/n = 0.5$. Under this condition, SPIM alone will predict the same steady-state relief for a 10 km$^2$ domain as a 1000 km$^2$ domain of the same horizontal shape, as illustrated below. LEMs utilizing SPIM often sidestep this problem with the use of a "hillslope diffusion" coefficient (e.g. Passalacqua
et al., 2006), a useful but rather poorly-constrained parameter that lumps together a wide range of processes (Fernandes and Dietrich, 1997). Alternatively, the problem can be sidestepped with an externally specified "hillslope critical length" (Goren et al., 2014a) that essentially specifies the location of channel heads. For example, the model simulations of Willett et al. (2014) employ the specific value of 500 m for hillslope critical length in their characterization of tendencies for drainage divide migration. The prediction of the hillslope diffusion coefficient and the location of channels are outstanding problems in the
field of geomorphology (Montgomery and Dietrich, 1988). The intrinsic nature of the SPIM model, however, is such that scale invariance persists for the case $m/n = 0.5$ at scales larger than a characteristic hillslope length scale, whether it be externally specified or computed from a diffusion coefficient.

The existence of scale invariance exemplifies an unrealistic aspect of SPIM, which we believe to be associated with its
omission of natural processes, such as abrasion due to sediment transport. Gilbert (1877) theorized two roles that sediment moving as bedload could play in bedrock incision, the first as an abrasive agent that incises the bed via collisions and the second as a protector that inhibits collisions of bedload on the bed. These observations have been implemented quantitatively

by many modelers (e.g., Sklar and Dietrich, 2001, Sklar and Dietrich, 2004; Sklar and Dietrich, 2006; Lamb et al., 2008; Zhang et al., 2015), some of whom have implemented them in LEMs (e.g. Gasparini et al., 2006, Gasparini et al., 2007). Egholm et al. (2013) have directly compared landscape models using SPIM on the one hand, and models using a saltation-abrasion model on the other hand. Here we shed light on an unrealistic behavior of SPIM with the goal of motivating the landscape evolution

community to develop more advanced treatments that better capture the underlying physics. A further goal is to emphasize the importance of scaling and non-dimensionalization in characterizing LEMs.

## 3 1D model: scale invariance and singularities

An LEM can be implemented using the following equation of mass conservation for rock/regolith subject to uplift and denudation:

$$\partial\eta/\partial t = v - E + D\nabla^2\eta \tag{2}$$

where $\eta$ = local landscape elevation, $t$ = time, $v$ = rock uplift rate and $D$ = hillslope diffusion coefficient. The term, $D\nabla^2\eta$, accounts for hillslope diffusion (Somfai and Sander, 1997; Banavar et al., 2001). The effect of diffusion is commonly neglected at coarse-grained resolution (Somfai and Sander, 1997; Banavar et al., 2001; Passalacqua et al., 2006), at which any resolved channels can be taken to be fluvially-dominated bedrock channels (Montgomery and Foufoula-Georgiou,1993). In our

analysis, we use Eq. (1) to specify the incision term in Eq. (2). It should be noted that SPIM refers to the incision in the direction normal to the bed, implying that there are both horizontal and vertical components of incision. In much of the literature using SPIM, however, the horizontal component is neglected in accordance with the original formulation of Howard and Kerby (1983), and incision is assumed to be purely vertical downward. Here we preserve this simplification in order to better understand the overall behavior of SPIM. Last, in correspondence with most 2D implementations of SPIM within LEM, we

neither resolve channels nor compute their hydraulic geometry in our 2D implementation. The focus of this paper is the most simplified form (e.g. (1)) of SPIM. This way we can analyze the most fundamental behavior of SPIM itself.

Equation (2) characterizes landscape evolution in 2D; i.e. elevation $\eta = \eta(x,y)$, where $x$ and $y$ are horizontal coordinates. It is useful for some purposes, however, to simplify Eq. (2) into a 1D form. Neglecting hillslope diffusion, the 1D conservation

equation is

$$\partial\eta/\partial t = v - KA^m(-\partial\eta/\partial l)^n \tag{3}$$

where $l$ = horizontal stream distance from the ridge, at which $l = 0$. It should be noted that the negative sign appears front of the term $\partial\eta/\partial l$ because $\partial\eta/\partial l$ is negative in the downstream direction, so that streambed slope, $S = -\partial\eta/\partial l$. In SPIM, slope $S$ is assumed to be positive. In order to solve Eq. (3), a relationship between $A$ and $l$ must be established. Here we assume a

generalized form of Hack's Law (Hack, 1957);

$$A = Cl^h \tag{4}$$

where $C$ and $h$ are positive values. Hack's Law assumes that upslope area increases with $l^h$. From empirical data, Hack found the exponent, $h$, to be ~1.67 (Hack, 1957).

Previous researchers have presented 1D analytical solutions for elevation profiles (Chase, 1992; Beaumont et al., 1992, Anderson, 1994; Kooi and Beaumont, 1994; Tucker and Slingerland, 1994; Kooi and Beaumont, 1996; Densmore et al., 1998; Willett, 1999; Whipple and Tucker, 1999; Willett, 2010). In their solutions, the effect of the horizontal scale, which in the 1D model we define as the total length of the stream profile, $L_{1D}$, was neither shown nor discussed. Previous studies that use Eq. (4) (Whipple and Tucker, 1999; Willett, 2010) involve nondimensionalization of both the horizontal and vertical coordinates by the total horizontal length of the profile, $L_{1D}$. As we show below, this step obscures the effect of the horizontal scale on the relief of the profile. In our study, we nondimensionalize the vertical coordinate, $\eta$, by a combination of $\upsilon$ and the acceleration of gravity, $g$. Our nondimensionalization of the coordinates is shown below.

$$\eta = \upsilon^2 g^{-1} \hat{\eta} \quad t = \upsilon g^{-1} \hat{t} \quad l = L_{1D} \hat{l} \tag{5}$$

Substituting Eq. (4) and Eq. (5) into Eq. (3) results in the following dimensionless conservation equation:

$$\partial \hat{\eta}/\partial \hat{t} = 1 - P_{1D}^{-n} \hat{l}^{hm} \left( -\partial \hat{\eta}/\partial \hat{l} \right)^n \tag{6}$$

where the dimensionless number $P_{1D}$, termed the 1D Pillsbury number herein for convenience, is given by the relation

$$P_{1D} = K^{-1/n} C^{-m/n} L_{1D}^{1-hm/n} \upsilon^{1/n-2} g \tag{7}$$

At steady-state, Eq. (6) becomes

$$P_{1D} = \hat{l}^{hm/n} \left( -\partial \hat{\eta}/\partial \hat{l} \right) \tag{8}$$

From this equation, we see that as we approach the ridge, i.e. $\hat{l} \to 0$, the slope term $\left( -\partial \hat{\eta}/\partial \hat{l} \right)$ always approaches infinity for positive values of $h$, $m$, and $n$.

The value of the 1D Pillsbury number $P_{1D}$ increases with stream profile length, $L_{1D}$ when $hm/n < 1$, is invariant to changes in $L_{1D}$ when $hm/n = 1$, and decreases with $L_{1D}$ when $hm/n > 1$. This can be further illustrated by integrating Eq. (8). To solve this first order differential equation, we need to specify a single boundary condition, shown below.

$$\hat{\eta}|_{\hat{l}=1} = 0 \tag{9}$$

This boundary condition sets the location and elevation of the outlet, where flow is allowed to exit the system. Integrating Eq. (8) yields

$$\hat{\eta} = \begin{cases} -P_{1D} \ln(\hat{l}) & \text{if } hm = n \\ (1 - hm/n)^{-1} P_{1D} \left( 1 - \hat{l}^{1-hm/n} \right) & \text{if } hm \neq n \end{cases} \tag{10}$$

The steady-state profiles defined by Eq. (10) are shown in Fig. 1. Inspecting Eq. (10), we see that elevation is infinite at the ridge ($l = 0$) when $hm/n \geq 1$, and elevation is finite when $hm/n < 1$. In addition, when $hm/n = 1$, $P_{1D}$ is no longer dependent on the horizontal scale, $L_{1D}$, and $\hat{\eta}$ is independent of the scale of the basin. Using the empirical value from Hack's original work

(1957), i.e. $h = 1.67$, the ratio, $m/n$, must take the value 0.6 for scale invariance. This ratio is within the range reported in the literature (Whipple and Tucker, 1999).

## 4 2D model: scale invariance

In 2D, the conservation equation using SPIM and neglecting hillslope diffusion can be written as

$$\partial \eta / \partial t = v - K A^m [(\partial \eta / \partial x)^2 + (\partial \eta / \partial y)^2]^{n/2} \tag{11}$$

To understand the behavior of Eq. (11) in response to scale, we need to use a dimensionless formulation in a fashion similar to the previous 1D analysis. Here, $L_{2D}$ denotes the horizontal length of the entire domain, which is taken to be square for convenience. For the 2D analysis, our nondimensionalization is

$$\eta = v^2 g^{-1} \hat{\eta} \quad t = v g^{-1} \hat{t} \quad A = L_{2D}^2 \hat{A} \quad x = L_{2D} \hat{x} \quad y = L_{2D} \hat{y} \tag{12}$$

The form of Eq. (11), in which $x$, $y$, and $A$ have been made dimensionless using the definitions shown in Eq. (12) is

$$\partial \hat{\eta} / \partial \hat{t} = 1 - P_{2D}^{-n} \hat{A}^m [(\partial \hat{\eta} / \partial \hat{x})^2 + (\partial \hat{\eta} / \partial \hat{y})^2]^{n/2} \tag{13}$$

where the dimensionless number $P_{2D}$, termed the 2D Pillsbury number is given as

$$P_{2D} = K^{-1/n} L_{2D}^{1-2m/n} v^{1/n-2} g \tag{14}$$

At steady-state, Eq. (13) becomes

$$P_{2D} = \hat{A}^{m/n} [(\partial \hat{\eta} / \partial \hat{x})^2 + (\partial \hat{\eta} / \partial \hat{y})^2]^{1/2} \tag{15}$$

The form of the parameter $P_{2D}$ specified by Eq. (14) is similar to the 1D form, Eq. (7), but is different due to the different dimensionality. The parameter, $P_{2D}$, scales with the relief of the landscape; as it increases, the slope term on the RHS of Eq. (15) also increases. The value of $P_{2D}$ increases with $L_{2D}$ for $m/n < 0.5$, remains constant with $L_{2D}$ for $m/n = 0.5$, and decreases with $L_{2D}$ for $m/n > 0.5$. For the ratio, $m/n = 0.5$, the exponent to which $L_{2D}$ is raised in Eq. (14) becomes zero, and the relief of the landscape becomes invariant to horizontal scale. When $m/n = 0.5$, the same steady-state solution to Eq. (15) prevails regardless of the value of $L_{2D}$. We note here that this scale-invariance, which is the key result of this paper, is intrinsic to the model itself and is not a function of the discretization scheme in used in implementing numerical solutions.

Our 2D model was solved using the following boundary conditions:

$$\eta|_{y=0} = 0 \tag{16}$$

$$\partial \eta / \partial y|_{y=L_{2D}} = 0 \tag{17}$$

$$\eta|_{x=0} = \eta|_{x=L_{2D}} \tag{18}$$

The bottom (outlet) side of the domain presented in Fig. 2 is fixed at the base level $\eta = 0$ m, corresponding to an open boundary where flow can exit the system while satisfying Eq. (16). The top side of the domain is designated as an impermeable boundary to flow, i.e. the drainage divide satisfies Eq. (17). Periodic boundary conditions satisfying Eq. (18) are applied at the left and right boundaries. Flow, slope, and drainage area are determined using the D8 flow algorithm, where flow follows the route of

steepest descent (O'Callaghan and Mark, 1984). The initial condition is a gently-sloped plane oriented towards the outlet with small random elevation perturbations.

For the results of Fig. 2, we use regular grids that contain $100^2$ cells. The number of cells is constant, regardless of the value of $L_{2D}$. This is in contrast to holding cell size constant and instead increasing the number of cells with $L_{2D}$. We argue that the former shows the fundamental behavior of SPIM, while the latter obscures this behavior due to the existence of slope and elevation singularities near the ridges in the landscape. The next sections show this singular behavior in the 2D numerical model.

Figure 2a shows steady-state solutions for $m/n = 0.5$ and two values of $L_{2D}$ using the same initial condition. At each corresponding grid cell between the two solutions, the slope, $S$, decreases as $L_{2D}$ increases. However, the relief structures of each landscape are identical. By relief structure, we are describing the elevation value at each corresponding grid cell in the two steady-state solutions. This is confirmed by nondimensionalizing the horizontal scale of landscape without adjusting the vertical scale (Fig. 2b). Using the same numerical methods and the parameters from Fig. 2a, the results of a similar analysis using different ratios $m/n = 0.4$, 0.5, and 0.6 are shown in Fig. 2c.

In Fig. 2c, the case of scale invariance can be seen when $m/n = 0.5$. For $m/n = 0.4$, the relief of the entire landscape increases with increasing $L_{2D}$, and for $m/n = 0.6$, the relief decreases with increasing $L_{2D}$. When $m/n \neq 0.5$, the landscapes do not exhibit scale invariance. However, the overall planform drainage network structure shows resemblance across scales. That is, the location of the major streams and rivers in the numerical grid are similarly organized. It should be noted that the landscapes are not identical. When the landscapes are shown in dimensional space, as shown in Fig. 2a, the landscapes appear to be quite different. In the case of Fig. 2b, however, the smaller landscape can be stretched horizontally to be precisely identical to the large one. The drainage network structure described above persists in each simulation due to the imprinting of the initial condition, which always consists of the same randomized perturbations.

**5 2D model: quasi-theoretical analysis of singular behavior**

Like the 1D model of Eq. (8), the 2D model, Eq. (15), has slope, $S$, approaching infinity as area, $A$, approaches zero at steady state. In contrast to the 1D model, however, general steady-state solutions for elevation in the 2D model, Eq. (15), cannot be determined analytically. However, the ratio, $m/n$, for which elevation singularities occur can be determined by analyzing the behavior of the 2D numerical model in close proximity to a ridge. Here, we first develop a quasi-theoretical treatment to study near-ridge behavior, and we then use it to infer singular behavior in the numerical model. Converting the coordinate system from Cartesian to a system that follows the streamwise direction, we rewrite Eq. (11) as

$$\partial \eta / \partial t = v - KA^m(-\partial \eta / \partial s)^n \tag{19}$$

where $s$ = distance along the path of steepest descent away from the ridge. From dimensional considerations, $A$ [$L^2$] must scale with $s^2$ [$L^2$] near the ridge ($s = 0$), and therefore,

$$A = \beta s^2 \text{ as } s \rightarrow 0 \tag{20}$$

where $\beta$ = scaling factor. For this analysis, our nondimensionalization is

$$\eta = v^2 g^{-1}\hat{\eta} \quad t = vg^{-1}\hat{t} \quad s = L_R\hat{s} \tag{21}$$

where $L_R$ = horizontal ridge scale. Near the ridge, Eq. (19) can be nondimensionalized into:

$$\partial\hat{\eta}/\partial\hat{t} = 1 - P_R^{-n}\hat{s}^{2m}(-\partial\hat{\eta}/\partial\hat{s})^n \tag{22}$$

where $P_R$ is another dimensionless Pillsbury number, here denoted as

$$P_R = K^{-1/n}\beta^{-m/n}L_{1D}^{1-2m/n}v^{1/n-2}g \tag{23}$$

At steady-state ($\partial\eta/\partial t = 0$), Eq. (22) becomes

$$P_R = \hat{s}^{2m/n}(-\partial\hat{\eta}/\partial\hat{s}) \tag{24}$$

From Eq. (24), we see that at the ridge ($\hat{s} = 0$), there is a singularity in slope, i.e. the slope, $(-\partial\hat{\eta}/\partial\hat{s})$, goes to infinity. Integration of Eq. (24) using the downstream boundary condition, $\hat{\eta}|_{\hat{s}=1} = 0$, allows for the delineation of the conditions for elevation singularities in the 2D model. The profile is given as

$$\hat{\eta} = \begin{cases} -P_R\ln(\hat{s}) & \text{if } 2m = n \\ (1 - 2m/n)^{-1}P_R(1 - \hat{s}^{1-2m/n}) & \text{if } 2m \neq n \end{cases} \tag{25}$$

Instead of the elevation singularity occurring when $hm/n \geq 1$ as seen in the 1D model, Eq. (10), this analysis for the 2D model shows an elevation singularity at the ridge when $m/n \geq 0.5$.

## 6 2D model: numerical analysis of singular behavior

In Fig. 3 and Fig. 4 we present results which serve to distinguish the fundamental behavior of SPIM from the numerical behavior associated with varying density of discretization. Fig. 3 and Fig. 4 each show nine steady state simulations, each using three values of $M^2$ and three values of $m/n$, i.e. 0.4, 0.5, and 0.6. In both figures, the number of cells is quadrupled from column to column. The leftmost column contains $40^2$ cells, the middle column contains $80^2$ cells, and the rightmost column contains $160^2$ cells. Figure 3 shows simulations where the horizontal length scale, $L_{2D}$, is held constant in all simulations. By increasing the number of cells, the grid size decreases. In all cases of $m/n$, the maximum relief increases with the number of cells. However, our quasi-theoretical analysis predicted the absence of an elevation singularity at the ridge for $m/n < 0.5$. To illustrate this point, we take a different approach, shown later in this section.

Figure 4 contains simulations where grid size is held constant at 125 m. Here, the horizontal length scale, $L_{2D}$, increases with the number of cells. In Fig. 4, the leftmost column contains $40^2$ cells with $L_{2D} = 5$ km, the middle column contains $80^2$ cells with $L_{2D} = 10$ km, and the rightmost column contains $160^2$ cells with $L_{2D} = 20$ km. Regardless of the $m/n$ ratio and whether $L_{2D}$ or grid size is kept constant, the maximum relief of the landscape increases as the number of cells increases. Relief increases

in both sets of simulations because with more grid cells, we are numerically sampling closer to ridges, and by sampling closer to ridges, we are resolving the ridge singularity at a finer scale. We emphasize, however, that the issue of dependence of the solution on grid size is separate from the issue of scale invariance for $m/n = 0.5$, the latter result being deduced from the governing equation itself (Eq. 15) before any discretization is implemented, and illustrated in Fig. 2c.

Our quasi-theoretical analysis infers the conditions for singular behavior in the 2D model. If elevation singularities exist, the model will not satisfy grid-invariance, causing the relief between the ridge and outlet to increase indefinitely as grid size decreases. In contrast, in simulations where singularities do not exist, the relief between the ridge and outlet can be expected to converge as the grid size decreases. In both cases, understanding ridge behavior in the 2D model requires studying solution behavior as grid size approaches zero.

We do this by extracting river profiles from 13 landscape simulations of different scales for each of three values of $m/n$, i.e. 0.4, 0.5 and 0.6. The largest simulation is for $L_{2D}^2 = 10^6$ km$^2$; simulations were also performed at progressively one order-or-magnitude less in area down to $L_{2D}^2 = 10^{-6}$ km$^2$. The number of grid cells, $M^2$, is held constant at $25^2$. In each simulation, then, the closest distance to the ridge that can be resolved is one grid cell, given by

$$\Delta l_i = 10^{(7-i)/2}/25 \, [km] \quad i = 1,2 \ldots 13 \tag{26}$$

From each of the simulations, we construct two synthetic river profiles, one that intersects the highest point of the basin divide (high profile) and one that intersects the lowest point of the basin divide (low profile). The choice of these two elevations was made so as to bracket the possible range of behavior; analogous results would be obtained from starting points along the basin divide at intermediate elevations. We use these synthetic profiles to characterize whether or not the numerical model is tending toward a singularity near ridges. We do this because the numerical model itself cannot directly capture singular behavior. We outline the details of the methodology for the high profile only, as the case of the low profile involves a transparent extension.

The 13 simulations result in 13 elevation profiles $\eta_i$, where $i = 1,2\ldots13$ each extending from $\Delta l_i$ (i.e. one grid point from the divide) to a downstream value $l_{Di}$ that is somewhat larger that the value $10^{3-(i/2)}$ km (because the down-channel path of steepest descent does not follow a straight line.). We assemble a synthetic channel profile, $\eta_S(l)$, from these as follows. The first leg of $\eta_S(l)$ is identical to $\eta_1(l)$, and extends from $l = \Delta l_1$ to $l_{D1}$. We extend the synthetic profile by translating the second profile upward until its elevation at its downstream point $l_{D2}$ matches with $\eta_S(l_{D2})$, as shown in Fig. 5a. The profile, $\eta_S(l)$, now extends from $\Delta l_2$ to $l_{D1}$. As shown in Fig. 5a, we repeat this process until all 13 profiles have been used to assemble the synthetic profile, which now extends from $\Delta l_{13}$ to $l_{D1}$.

25

30

This procedure results in a high synthetic profile encompassing all thirteen profiles (circles) and in a low synthetic profile (crosses) (Fig. 5b). 1D analytical solutions, Eq. (10), are then fitted to the profiles of the 2D simulations using the 1D Pillsbury

number, $P_{ID}$, as a fitting parameter. To account for the difference in dimensionality, the 1D steady-state profiles with $hm/n =$ 0.8, 1.0, and 1.2 are fitted to the 2D data for $m/n = 0.4$, 0.5, and 0.6, respectively. The scatter in the synthetic profile is due to the randomness in the pathway, as dictated by the initial conditions.

Figure 5b shows good fit between the 2D results and the corresponding 1D steady-state profiles. This allows us to make inferences concerning asymptotic behavior at a ridge. The analytical curves for elevation that best fit the 2D data for $m/n < 0.5$ converge to finite values as $l$ approaches 0 and infinity for $m/n \geq 0.5$. While these results do not constitute analytical proof of this asymptotic behavior, they provide compelling evidence for it.

## 7 Scale behavior in other landscape evolution models

We offer here an example of a landscape model that does not necessarily satisfy horizontal scale invariance, i.e. that of Gasparini et al. (2007). They incorporate the formulation of Sklar and Dietrich (2004) for bedrock abrasion due to wear in their model. The rate of erosion $E$ is given as

$$E = K_{GA}(1 - Q_s/Q_t)Q_s/W \tag{27}$$

where $K_{GA}$ = abrasion coefficient, $Q_s$ = bedload sediment flux, $W$ = channel width, and $Q_t$ = bedload transport capacity.

Gasparini et al (2007) use the following relation for $Q_t$.

$$Q_t = K_t A^{m_t} S^{n_t} \tag{28}$$

where $K_t$ is a transport constant, and $m_t$ and $n_t$ are exponents. At steady state, the total sediment flux at any point in the landscape must equal the production rate of sediment due to rock uplift:

$$Q_S = K_B A \upsilon \tag{29}$$

where $K_B$ is the fraction of sediment produced that contributes to bedload (the remainder being moved out of the system as washload). For channel width, they use a relation of the form

$$W = k_w Q^b \tag{30}$$

where $Q$ = water flow discharge, $k_w$ = hydraulic geometry constant, $b$ = hydraulic geometry exponent (e.g. Finnegan et al., 2005). The value of $b$ has been found to vary between 0.3 and 0.5 for bedrock rivers (Whipple 2004); Gasparini et al. (2007)

use $b = 0.5$ in their model. They also estimate discharge as an effective precipitation rate, $k_q$, multiplied by a drainage area to the power of $c$, where $c \leq 1$

$$Q = k_q A^c \tag{31}$$

The resulting relation for steady-state slope is:

$$S = [(\partial \eta/\partial x)^2 + (\partial \eta/\partial y)^2]^{1/2} = (K_B K_t^{-1} \upsilon A^{1-m_t})^{1/n_t} \left(1 - k_q^b k_w K_B^{-1} K_{GA}^{-1} A^{bc-1}\right)^{-1/n_t} \tag{32}$$

Using the nondimensionalization terms from (12), we nondimensionalize (32) to

$$[(\partial\hat\eta/\partial\hat x)^2 + (\partial\hat\eta/\partial\hat y)^2]^{1/2} = P_{G1}\hat A^{1/n_t - m_t/n_t}\left(1 - P_{G2}\hat A^{bc-1}\right)^{-1/n_t} \tag{33}$$

where $P_{G1}$ and $P_{G2}$ are two dimensionless Pillsbury numbers:

$$P_{G1} = v^{-2}g\left(K_B K_t^{-1} v L_{2D}^{2-2m_t+n_t}\right)^{1/n_t} \tag{34}$$

$$P_{G2} = K_B^{-1} K_{GA}^{-1} L_{2D}^{2bc-2} \tag{35}$$

Horizontal scale invariance results only when both dimensionless numbers are independent of the horizontal length scale, $L_{2D}$. Gasparini et al. (2007) use $m_t = 1.5$ and $n_t = 1.0$. This parameter does indeed make the exponent, $2 - 2m_t + n_t$, equal to zero, so that $P_{G1}$ is independent of $L_{2D}$. The parameter $P_{G2}$ is invariant to the horizontal scale when the product of $b$ and $c$ is equal to one. However, realistic values of $b$ are between 0.3 and 0.5 (Whipple, 2004), and value of $c$ is less than or equal to 1. This means that the maximum value of $bc$ is 0.5. It follows that $P_{G2}$ is not independent of the horizontal scale, and that the model

of Gasparini et al. (2007) does not satisfy horizontal scale invariance.

## 8. Sensitivity of relief to hillslope length and profile length

In the river profiles of Fig. 1 and Fig. 5b, we see that a sizable proportion of the relief is confined to the headwaters, i.e. near a ridge. In our 1D model, for $hm/n \geq 1$, ridge elevation is infinite, thus formally implying infinite relief. This problem has been sidestepped by introducing a critical hillslope length $l_c$, upstream of which it is assumed that there is no channel (e.g. Goren et

al., 2014a). This point may be thought of as loosely corresponding to the channel-hillslope transition in the slope-area relation discussed by Montgomery and Dietrich (1988) and Montgomery and Dietrich (1992). Here, then, we let the hillslope zone cover the range $0 \leq l \leq l_c$, where $l_c$ is an appropriately small fraction of profile length $L_{1D}$. Modifying Eq. (10) accordingly, we can determine the total relief, $R$, of the channel profile as follows;

$$\hat R = \begin{cases} -P_{1D}\ln(\hat l_c) & \text{if } hm = n \\ (1 - hm/n)^{-1}P_{1D}\left(1 - \hat l_c^{1-hm/n}\right) & \text{if } hm \neq n \end{cases} \tag{36}$$

where

$$R = v^2 g^{-1}\hat R \quad l_c = L_{1D}\hat l_c \tag{37}$$

We remind the reader that according to Eq. (7),

$$P_{1D} \sim L_{1D}^{1-hm/n} \tag{38}$$

We now consider the scale-invariant case, $hm/n = 1$, and inquire as to how the relief of the basin might change. Increasing $L_{1D}$ does not increase relief, because the parameter $P_{1D} \sim L_{1D}^0$. It is thus seen from Eqs. (36) and (37) that relief can be increased only by decreasing $\hat l_c$. But from (36), $\hat R \to \infty$ as $\hat l_c \to 0$. It follows that relief is extremely sensitive to the choice of $\hat l_c$. Based on our previous analysis, we expect that this result carries over to the case $m/n = 0.5$ for the 2D model.

We next provide an example illustrating the dependence of relief on hillslope length and profile length when $hm/n \neq 1$. Specifically, we consider the case $hm/n = 0.9$, with a dimensionless hillslope length $\hat{l}_c = 0.01$. According to Eq. (36), a halving of $\hat{l}_c$ to 0.005 increases the relief by 11.4 percent. In order to achieve the same increase in relief by changing profile length $L_{1D}$ while holding $\hat{l}_c$ constant, $L_{1D}$ would have to be increased by 196%. It is thus seen that relief of the channel profile can be more sensitive to a relative change in dimensionless critical channel length than it is to a relative change in horizontal scale.

## 9 Discussion and conclusion

Our 1D analytical solutions, Eq. (10) and Fig. 1, characterize the scale behavior of 1D SPIM, with horizontal scale invariance satisfied when $hm/n = 1.0$. Our 2D numerical solutions shown in Fig. 2 illustrate our analytical result that 2D SPIM shows horizontal scale invariance when $m/n = 0.5$. That is, 2D models using SPIM with $m/n = 0.5$ show the same relief structure regardless of the horizontal scale. This scale invariance has been previously demonstrated for neither the 1D nor the 2D SPIM model. Our result calls into question the common usage of the ratio $m/n = 0.5$ in landscape evolution models (Gasparini et al., 2006).For example, the Python-based landscape modelling environment, Landlab (Hobley et al., 2017) offers a default $m/n$ ratio of 0.5. Our result also motivates further investigation as to why analysis of field data commonly yields values of $m/n \sim$ 0.5 (e.g. Snyder et al., 2000). It should be noted that local empirical measurements indicating $m/n = 0.5$ do not necessarily mean $m/n = 0.5$ should be used as a universal ratio in SPIM. Gasparini and Brandon (2011) used multiple incision laws, other than SPIM, to simulate steady state landscapes, and were able to fit $E$, $A$, and $S$ in Eq. (1) to find empirical values of $m'$ and $n'$ (prime denotes an empirical value). They found that the ratio of $m'/n'$ was sensitive to the incision model's parameters as well as the rock uplift pattern in each landscape. This implies that both $m'$ and $n'$ have dependency on landscapes properties and are not universal from landscape to landscape.

In addition to the horizontal scale invariant case $m/n = 0.5$ for the 2D SPIM model, we also emphasize the relationship between the steady state landscape relief and horizontal when $m/n \neq 0.5$. Eq. (14) and Eq. (15) and the results in Fig. 2c show that the relief structure of the landscape scales with $P_{2D}$. Within $P_{2D}$, the horizontal length scale term is $L_{2D}^{1-2m/n}$. For the $m/n$ ratio range 0.35 to 0.6 (Whipple and Tucker 1999), the corresponding exponent range in the horizontal length scale term is - 0.2 to 0.3. This means that over the stated range of $m/n$, the relief structure has a weak dependence on the horizontal length scale. For $m/n < 0.5$, relief weakly increases with horizontal scale. For $m/n > 0.5$, relief weakly but unrealistically decreases with horizontal scale. The underlying physics of channel and hillslope processes that might dictate such behaviour are, at present, unhelpfully opaque. In natural systems, larger landscapes would yield longer rivers. Since elevation monotonically increases with upstream distance, one would expect relief to increase with horizontal scale. The results of SPIM, where $m/n \geq 0.5$, clearly contradict this intuitive understanding.

Our work neglects the effect of hillslope diffusion because our intent is to study the behavior of SPIM itself. Without hillslope diffusion, SPIM causes singular behavior at ridges in both the 1D and 2D formulation. Indeed, both the 1D and 2D models exhibit singularities in slope at ridges for all $hm/n$ ratios (1D) and all $m/n$ ratios (2D). For $hm/n \geq 1$ (1D) and $m/n \geq 0.5$ (2D), the models exhibit singular behavior in elevation at ridges as well. When relief is limited by a hillslope length $l_c$, elevation and slope do indeed reach finite values at the channel heads, but the effects of the singularity persist. For example, for the case $hm/n \geq 1$ in the 1D model, relief approaches infinity as hillslope length approaches zero. Our analysis of ridge singularities in SPIM shows that the choice of hillslope parameterization plays a key role in determining the relief of natural landscapes.

Numerical solutions of the 2D model indicate that it cannot be grid-invariant for $m/n \geq 0.5$. In the absence of hillslope diffusion, ridges reach infinite elevation as grid size becomes vanishingly small. This result underlines the critical role of hillslope diffusion in obtaining meaningful results from the 2D model. Field estimates of hillslope diffusion have been obtained at the hillslope scale, but there are unanswered questions about their application to large-scale models (Fernandes and Dietrich, 1997). Our results suggest that for the ratio, $m/n < 0.5$, there are steady-state grid-invariant solutions. However, the grid size below which grid-invariance is realized may be so small, e.g. sub-meter scale, that the validity of Eq. (1) is called into question. Issues with SPIM when used at large scales include the following. Studies commonly neglect the effect of hillslope diffusion when the scale of the grid is larger than the hillslope scale (Somfai and Sander, 1997; Banavar et al., 2001; Passalacqua et al., 2006). At coarse-grained scales, increasing the size of the numerical domain, while keeping the number of cells constant, will result in the behavior shown in Fig. 2. In Fig. 4 we see that adding more cells to compensate for the increase in size of the domain, such that the grid size remains constant, produces heavily biased (i.e. ever more singular) behavior near the ridges.

Our analysis illustrates that SPIM has two important limitations; a) unrealistic scale invariance when $m/n$ takes the commonly-used value 0.5, so that a 10 km$^2$ basin has identical relief to a 1000 km$^2$ basin, and b) singular behavior near the ridges for $m/n \geq 0.5$ that makes maximum relief entirely and unrealistically dependent on grid size. SPIM has been shown to be of considerable use in the study of the general behavior of landscapes (e.g., Howard, 1994; Howard et al., 1994). We believe, however, that the time has come to move on to more sophisticated models. While scientific questions remain that can be answered with the stream power incision model, there are many more questions that require a more advanced formulation (e.g. Gasparini et al., 2007 Crosby et al., 2007, Egholm et al., 2013). The development of alternative, more physically-based models for incision (e.g. Sklar and Dietrich, 2004; Lague, 2014; Zhang et al., 2015) and their application to landscape evolution (e.g. Davy and Lague, 2009; Gasparini et al., 2006, 2007) offer exciting prospects for the future.

**10 Notation**

$A$        upslope drainage area [L$^2$]

$b$        exponent defining relation between channel width and flow discharge (Gasparini et al., 2007) [-]

| | | |
|---|---|---|
| $B$ | profile width [L] | |
| $c$ | exponent defining relation between flow discharge and drainage area (Gasparini et al., 2007) [-] | |
| $C$ | Hack's law constant [L$^{2-h}$] | |
| $D$ | hillslope diffusion coefficient [L$^2$/T] | |
| 5 $E$ | local erosion rate [L/T] | |
| $g$ | acceleration of gravity [L/T$^2$] | |
| $h$ | Hack's law exponent [-] | |
| $K$ | erodibility coefficient [L$^{(1-2m)}$/T] | |
| $K_B$ | fraction of sediment produced that contributes to bedload (Gasparini et al., 2007) [-] | |
| 10 $K_{GA}$ | constant defining relation for the general abrasion model (Gasparini et al., 2007) [L$^{-1}$] | |
| $k_q$ | effective precipitation rate (Gasparini et al., 2007) [L$^{(3-2c)}$/T] | |
| $K_t$ | constant defining relation for bedload transport capacity (Gasparini et al., 2007) [L$^{(3-2mt)}$/T] | |
| $k_w$ | constant defining relationship between channel width and flow discharge (Gasparini et al., 2007) [L$^{(1-3b)}$T$^b$] | |
| $i$ | index denoting the profile, 1,2…13 [-] | |
| 15 $l$ | horizontal distance from the ridge in the 1D profile [L] | |
| $\hat{l}$ | dimensionless horizontal distance from the ridge in the 1D profile, $l/L_{1D}$ [-] | |
| $l_c$ | critical hillslope length[L] | |
| $\hat{l}_c$ | dimensionless critical hillslope length, $l_c/L_{1D}$ [-]$l_{Di}$   total length of profile, $i$ [L] | |
| $l_i$ | horizontal distance from the ridge of profile, $i$ [L] | |
| 20 $L_{1D}$ | horizontal length scale, profile length [L] | |
| $L_{2D}$ | horizontal length scale, basin size [L] | |
| $L_R$ | horizontal length scale, ridge [L] | |
| $m$ | exponent above $A$ in SPIM [-] | |
| $m_t$ | exponent above $A$ in sediment transport capacity equation (Gasparini et al., 2007) [-] | |
| 25 $M^2$ | number of numerical cells [cells$^2$] | |
| $n$ | exponent above $S$ in SPIM [-] | |
| $n_t$ | exponent above $S$ in sediment transport capacity equation (Gasparini et al., 2007) [-] | |
| $P_{1D}$ | Pillsbury number for the 1D analysis [-] | |
| $P_{2D}$ | Pillsbury number for the 2D analysis [-] | |
| 30 $P_{G1}$ | first Pillsbury number for the Gasparini et at. (2007) analysis [-] | |
| $P_{G2}$ | second Pillsbury number for the Gasparini et at. (2007) analysis [-] | |
| $P_R$ | Pillsbury number for the 2D ridge analysis [-] | |
| $Q_s$ | bedload sediment flux [L$^3$/T] | |

| | | |
|---|---|---|
| $Q_t$ | bedload transport capacity [L$^3$/T] | |
| $R$ | total relief of the channel profile [L] | |
| $\hat{R}$ | dimensionless total relief, $Rg/v^2$ [-] | |
| $s$ | distance from the ridge [L] | |
| 5 $\hat{s}$ | dimensionless distance from the ridge, $s/L_R$ [-] | |
| $S$ | stream gradient [-] | |
| $t$ | time [T] | |
| $\hat{t}$ | dimensionless time, $tg/v$ [-] | |
| $W$ | channel width [L] | |
| 10 $x$ | horizontal coordinate orthogonal to $y$ [L] | |
| $\hat{x}$ | dimensionless horizontal coordinate, $x/L_{2D}$ [-] | |
| $y$ | horizontal coordinate orthogonal to $x$ [L] | |
| $\hat{y}$ | dimensionless horizontal coordinate, $y/L_{2D}$ [-] | |
| $\beta$ | ridge scaling constant [-] | |
| 15 $\Delta l_i$ | grid size for profile, $i$ [L] | |
| $\eta$ | elevation [L] | |
| $\hat{\eta}$ | dimensionless elevation, $\eta g/v^2$ [-] | |
| $\eta_i$ | elevation of profile, $i$ [L] | |
| $\eta_S$ | elevation of synthetic profile [L] | |
| 20 $v$ | uplift rate [L/T] | |

**Acknowledgements**

This material is based upon work supported by the US Army Research Office under Grant No. W911NF-12-R-0012 and by the National Science Foundation Graduate Research fellowship under Grant No. DGE-1144245.

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

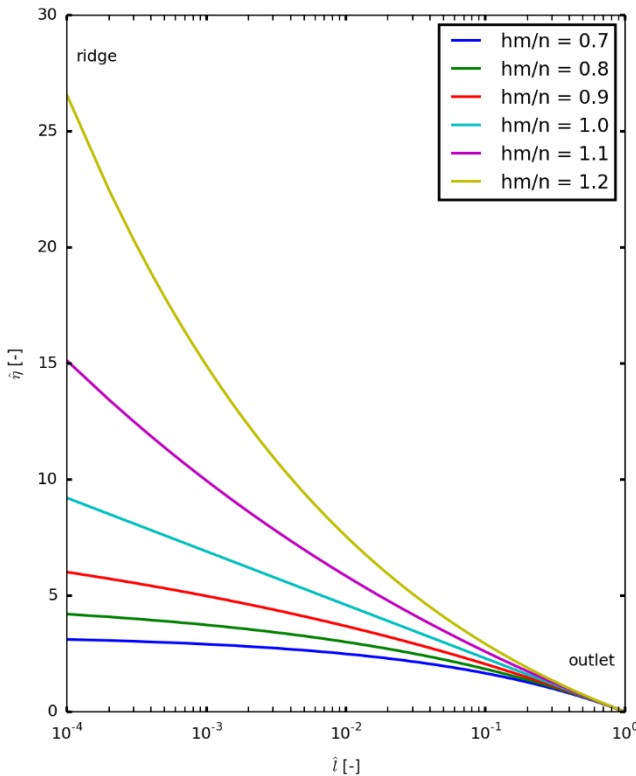

**Figure 1: 1D analytical dimensionless solutions for elevation profiles at steady-state equilibrium over a range of ratios *hm*/*n* (Hack's Law) =0.7, 0.8, 0.9, 1.0, 1.1, and 1.2 and *P*$_{ID}$ = 1.0.**

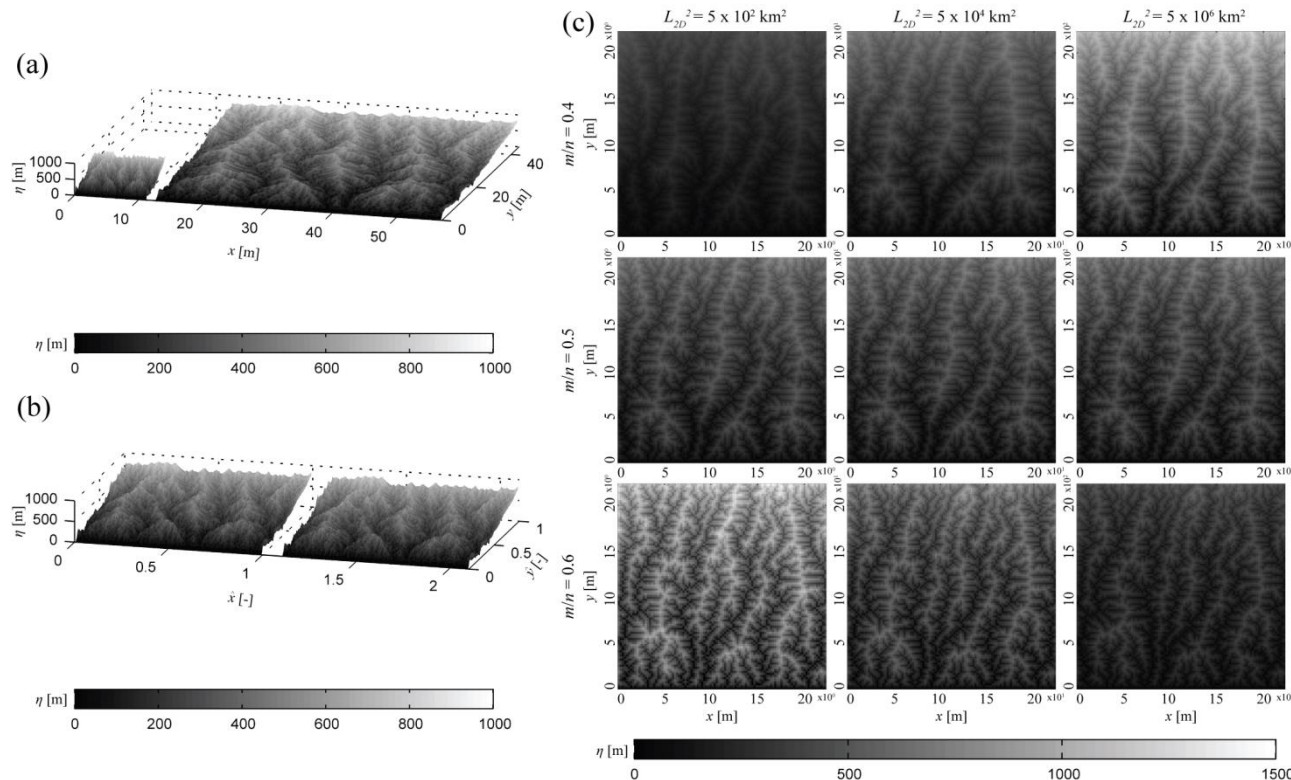

**Figure 2: (a) 2D numerical landscapes at steady-state using a ratio of *m/n* = 0.5, *n* = 1.0, *v* = 4 mm/yr, *K* = 2.83x10⁻¹¹ s⁻¹, *M²* = 100²**
**cells, and *L₂D²* = 125 km² and 2000 km². For each case, the 2D Pillsbury number was the same, 2.73x10²¹. (b) Results of (a) expressed**
**in terms of dimensionless horizontal scale. Each basin is made dimensionless by its basin size, *L₂D*. (c) Nine 2D numerical simulations**
**at dynamic equilibrium for three different values of *L₂D* and three different values of *m/n*. The value of *K* has been chosen to be**
**different for each value of *m/n* for clarity in the figures. From left to right, the *L₂D²* = 5x10² km², 5x10⁴ km², and 5x10⁶ km². To make**
**the relief of the landscapes comparable, the 2D Pillsbury number, *P₂D*, is set to 2.73x10²¹ for solutions of all *m/n* ratios with *L₂D²* =**
**5x10² km². To achieve this for *v* = 4 mm/yr, *K* = 2.10x10⁻¹⁰ m⁰·²/s, 2.83x10⁻¹¹ s⁻¹, and 3.82x10⁻¹² m⁻⁰·²/s for *m/n* = 0.4, 0.5, and 0.6,**
**respectively.**

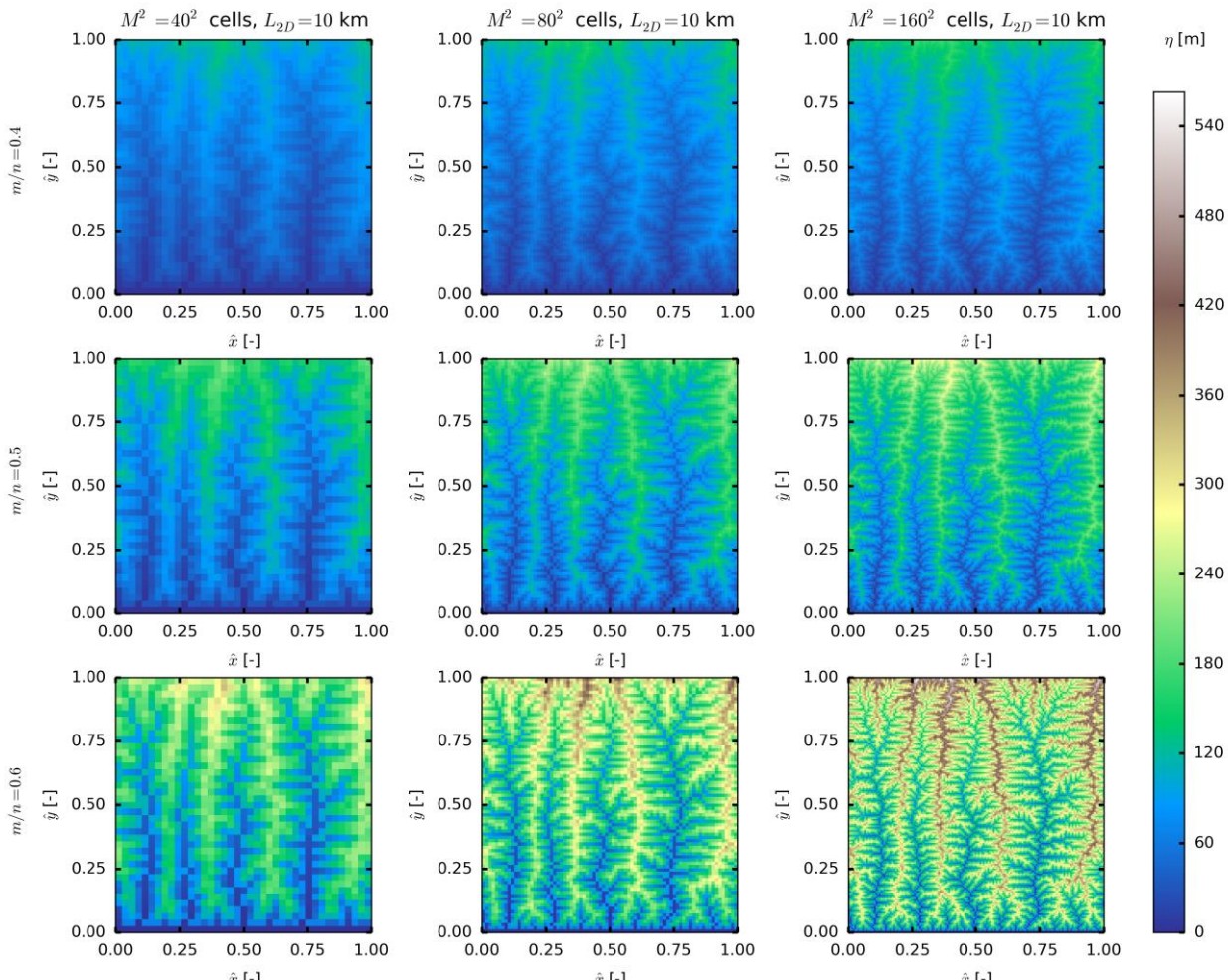

**Figure 3: Nine 2D numerical simulations at steady state for three different values of $M^2$ and three different values of $m/n$. In this figure, the horizontal scale is kept constant; $L_{2D}$ = 10 km for all solutions. The value of $K$ has been chosen to be different for each value of $m/n$ for clarity in the figures. From left to right, the number of cells $M^2 = 40^2$, $80^2$, and $160^2$. To make the relief of the landscapes comparable, the 2D Pillsbury number, $P_{2D}$, is set to $3.10 \times 10^{23}$ for solutions of all $m/n$ ratios with $L_{2D}$ = 10 km. To achieve this for $\upsilon$ = 1 mm/yr, $K = 6.31 \times 10^{-12}$ m$^{0.2}$/s, $1.00 \times 10^{-12}$ s$^{-1}$, and $1.58 \times 10^{-13}$ m$^{-0.2}$/s for $m/n$ = 0.4, 0.5, and 0.6, respectively. Relief increases with the number of cells because the ridge singularity is resolved at finer resolution.**

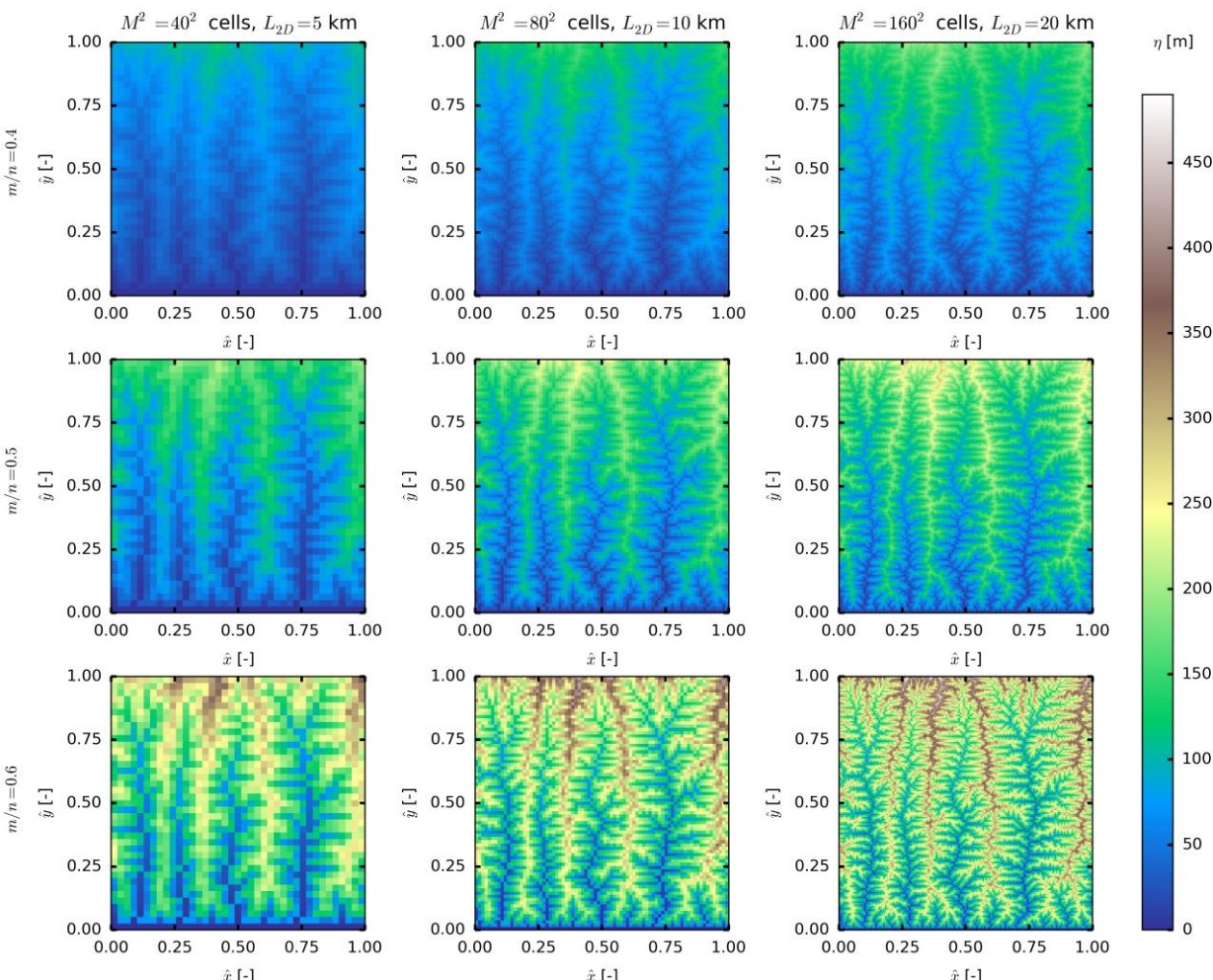

**Figure 4: Nine 2D numerical simulations at steady state for three different values of $M^2$ and three different values of $m/n$. In this figure, the grid size, $L_{2D}/M = 125$ m, is used in all the solutions. The value of $K$ has been chosen to be different for each value of $m/n$ for clarity in the figures. From left to right, the number of cells $M^2 = 40^2$, $80^2$, and $160^2$. To make the relief of the landscapes comparable, the 2D Pillsbury number, $P_{2D}$, is set to $3.10 \times 10^{23}$ for solutions of all $m/n$ ratios with $L_{2D} = 10$ km. To achieve this for $v = 1$ mm/yr, $K = 6.31 \times 10^{-12}$ m$^{0.2}$/s, $1.00 \times 10^{-12}$ s$^{-1}$, and $1.58 \times 10^{-13}$ m$^{-0.2}$/s for $m/n = 0.4$, 0.5, and 0.6, respectively. Like Figure 4, relief increases with the number of cells because the ridge singularity is resolved at a finer resolution.**

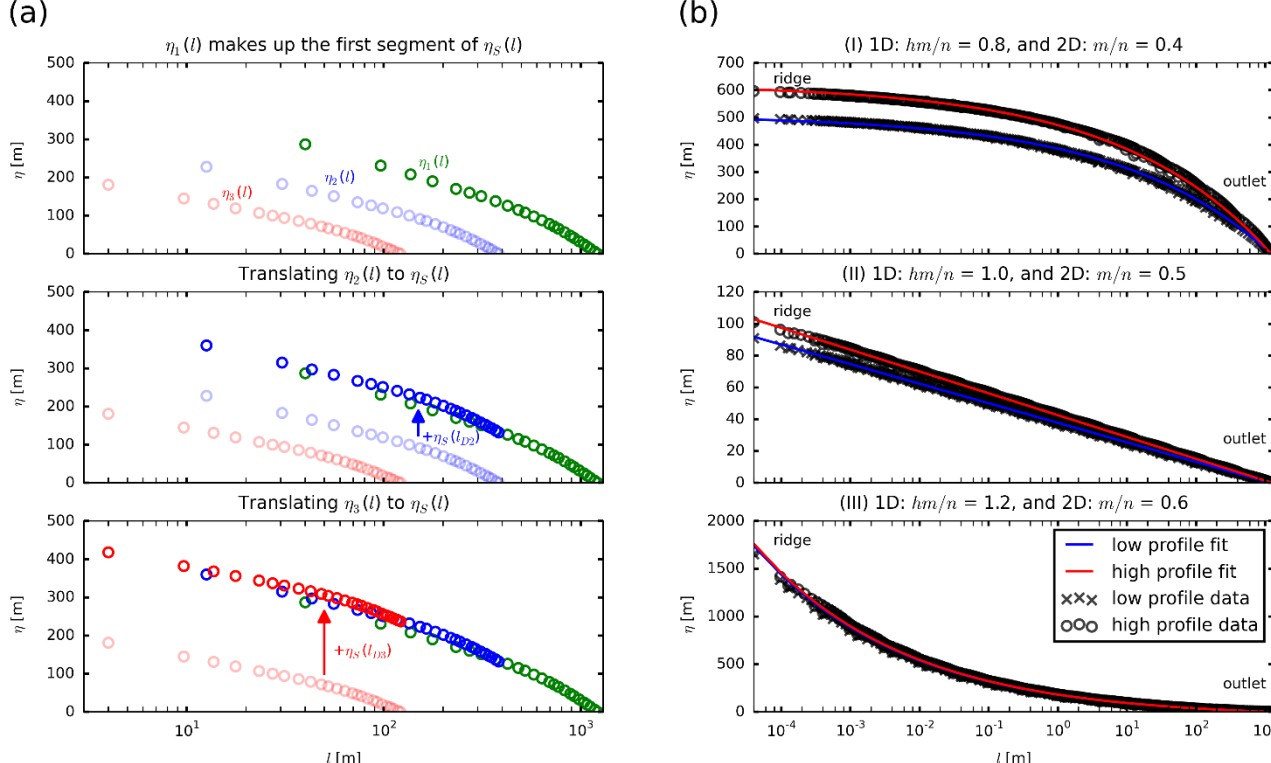

**Figure 5: (a)** Construction of the synthetic profile, $\eta_S(l)$. **The opaque points represent the synthetic profile, and the transparent points represent the untranslated profiles. The green points represent the profile for** $i = 1$**, blue represent** $i = 2$**, and red represent** $i = 3$**. After** $\eta_{13}(l)$ **has been utilized in** $\eta_S(l)$**, the synthetic profile is complete. (b) 1D steady-state equilibrium analytical solutions fitted to 2D numerical results using** $P_{1D}$**. Each** $m/n$ **ratio contains two profiles, one generated from a flow path from the highest point on the ridge corresponding to the basin divide (HP) and one from the lowest point on the basin divide (LP). The circles (HP) and crosses (LP) represent the 2D model data, and the red (HP) and blue (LP) line represent the 1D analytical model. For each** $m/n$ **ratio,** $v = 3$ **mm/yr,** $M^2 = 25^2$ **cells,** $n = 1.0$**, and** $L_{2D}^2 = 10^{-6}$ **km$^2$ to** $10^6$ **km$^2$. (I) Using** $K = 5.00 \times 10^{-12}$ **m$^{0.2}$/s,** $m/n = 0.4$ **(2D), and** $hm/n = 0.8$ **(1D),** $P_{1D}$ **= 6.45$\times$10$^{21}$ (LP) and** $P_{1D} = 7.89 \times 10^{21}$ **(HP). (II) Using** $K = 2.83 \times 10^{-11}$ **s$^{-1}$,** $m/n = 0.5$ **(2D), and** $hm/n = 1.0$ **(1D),** $P_{1D} = 5.79 \times 10^{21}$ **(LP) and** $P_{1D} = 6.47 \times 10^{21}$ **(HP). (III) Using** $K = 3.82 \times 10^{-12}$ **m$^{-0.2}$/s,** $m/n = 0.6$ **(2D), and** $hm/n = 1.2$ **(1D),** $P_{1D} = 2.13 \times 10^{23}$ **(LP) and** $P_{1D} = 2.15 \times 10^{23}$ **(HP).**