# Peer review of "Landscape evolution models using the stream power incision model show unrealistic behavior when $m/n$ equals 0.5"

_Earth Surface Dynamics, 2017_

## Referee Comment (RC1) · Anonymous Referee #1 · 11 May 2017

This paper shows that if the equation

del z / del t = U – K * A^m * S^n

is made dimensionless, the parameters form a non-dimensional group that includes Lˆ(2m-n), where L is a horizontal length. Thus, if 2m = n, the dimensionless governing equation is independent of L, and topographic surfaces that satisfy that equation do not depend on horizontal length. The authors demonstrate this result numerically by solving the dimensional equation on identical grids with the same initial surface but different delta x and delta y and obtaining horizontally identical solutions. They also point out that near drainage divides, where A is small, the second term on the RHS becomes small, and elevations in the solution become large. They interpret these

results as evidence that the stream-power term is an inadequate model of bedrock channel incision.

The singularity at A = 0 is well known. It is a part of the solution that exists on paper but is never realized in nature because other mechanisms dominate erosion near drainage divides, where drainage area is small. The authors already seem to consider this a secondary point – they don't mention it in the abstract – so removing it would not change the paper much.

The special mathematical case for 2m=n is interesting, and the thorough analysis presented in the paper could form an important part of a more general study of scaling in landscape evolution models. However, I am not convinced that a paper that presents only this result can stand on its own. The demonstrated scale invariance occurs in a model from which terms that impart scale dependence have been omitted. One such term is the diffusion term in equation 2 (which should be positive). The authors argue that hillslope diffusion "operates only at small scales". Sure, but might that not contradict the conclusion that "the steady-state landscape for a 1 mˆ2 domain can be stretched so that it is identical to the corresponding landscape for a 100 kmˆ2 domain"? The authors also do not consider channel width, another potential source of scale dependence.

I appreciate what the authors are trying to do: discovering flaws in widely used models is one way that science advances. But they seem to construe their discovery as evidence that the entire community is asleep at the wheel, and I don't think that is true. My impression is that most researchers who use the stream power model understand that it is potentially relevant and potentially useful only at scales where erosional channels form, and that the form of the model written in terms of drainage area and slope is a convenient geometric simplification that is useful only under certain conditions. The fact that this simplification gives rise to scale invariance with a particular combination of parameters is indeed an odd quirk – one that is probably worthy of a cautionary tale – but it doesn't mean that the underlying arguments for relating incision rate to
**ESurfD**

Interactive
comment

drainage area and slope are fundamentally flawed. The version of the stream power model presented in this paper certainly has substantial limitations, and discussions of its shortcomings – as well as proposed improvements – abound in the literature.

I see two ways in which the authors could potentially use their analysis to contribute to those discussions. First, perhaps they could show more clearly how scale-invariant models would lead researchers to draw incorrect conclusions about drainage basins, even if those researchers are aware of the limitations of the stream power model as a process law. Second, they might consider whether the particular shortcomings they document offer any insights into how a better model of river incision could be con-structed – and whether any of the proposed improvements to the stream power law avoid these problems.

---

## Referee Comment (RC2) · Anonymous Referee #2 · 27 May 2017

This paper presents a call to arms, urging landscape evolution modelers who use the stream power incision model (SPIM) to "move on to more sophisticated models", which better represent the physical mechanisms responsible for river erosion of bedrock, such as abrasion by sediment. The argument rests primarily on the finding of scale invariant solutions when the SPIM exponent ratio m/n = 0.5, for the case where the commonly-used hillslope "diffusion" term is omitted. Overall, the paper is well written, and the analysis is clearly presented.

While I am sympathetic to the stated goals of this work, I worry that, ironically, this paper may have the opposite impact by focusing so narrowly on a rather anecdotal result. The model behavior described here will rarely occur in model studies because

modelers typically use other m/n ratios, or hillslope diffusion terms, minimum hillslope lengths or other model components that avoid this result. Will this finding convince anyone to abandon what has become the standard model for the advective component of landscape evolution modeling? I doubt it. More likely, this result will be cited perfunctorily in statements that acknowledge the limitations of the SPIM to assuage the concerns of reviewers who might prefer the use of "more sophisticated models".

I agree with the suggestions of the first reviewer for how this work could be extended in constructive ways. For example, can scale analysis be used to identify when the SPIM may lead to incorrect interpretations, or test the validity of divergent model outcomes, such as the findings of Egholm et al. (2013) who directly compared the SPIM with a bedload abrasion incision model? Alternatively, this paper might stand on its own as a technical note, if motivated more narrowly by a problem this result helps to solve.

Egholm, David L., Mads F. Knudsen, and Mike Sandiford. "Lifespan of mountain ranges scaled by feedbacks between landsliding and erosion by rivers." Nature 498.7455 (2013): 475-478.

---

## Author Comment (AC1) · 23 Jun 2017

We thank the reviewer for their thoughtful comments and suggestions. From the review's introduction, we believe the reviewer has a good understanding of our main results and our motivation for writing this paper.

**Reviewer 1**: *"The singularity at A = 0 is well known."*

There are two issues here: elevation singularity and slope singularity at the ridges.

[Figure]

We agree that it is readily seen that there is a slope singularity at the ridge at steady state when drainage area goes to zero. However, without integrating the conservation equation at steady state, the existence of an elevation singularity at the ridges for $m/n \geq 0.5$, and its absence for $m/n < 0.5$ cannot be easily deduced. This is especially true in the 2D model, which has no analytical solution. Within the literature, there has been little discussion specifically oriented to the singular behavior of SPIM in regard to slope at the ridge. To our knowledge, the presence or absence of the elevation singularity (according to the value of $m/n$) in 2D models has never been shown in the literature.

**Reviewer 1**: *"It is a part of the solution that exists on paper but is never realized in nature because other mechanisms dominate erosion near drainage divides, where drainage area is small."*

We agree with the comment. But our goal is not to understand how SPIM performs in conjunction with other mechanisms, but rather to see how SPIM itself performs. For this reason, we did not include hillslope diffusion in our model. There are two additional issues. 1: When the using a grid size in the 2D model that is larger than the hillslope length scale, hillslope diffusion has little to no effect on the landscapes relief. 2: When the basin is sufficiently large compared to, e.g. the scale of hillslope diffusion, unrealistic horizontal scale invariance prevails at all scales larger than that of hillslope diffusion when $m/n = 0.5$.

**Reviewer 1**: *"The authors already seem to consider this a secondary point – they don't mention it in the abstract – so removing it would not change the paper much."*

We do not agree with this statement; the singular behavior is an integral part of this paper. It is our belief that the singular behavior is an important for demonstrating some

of the pitfalls of modeling landscapes with SPIM. The reviewer makes good points here, and we thus propose to expand our discussion on the singularity and focus on explaining its importance.

**Reviewer 1**: *"The special mathematical case for 2m=n is interesting, and the thorough analysis presented in the paper could form an important part of a more general study of scaling in landscape evolution models. However, I am not convinced that a paper that presents only this result can stand on its own."*

Our paper does indeed focus on the scale invariant case, as we believe it is the most interesting and surprising part of the analysis, and corresponds to the most commonly used value of $m/n$. However, looking at Figure 2, we not only present the scale invariant case of $m/n = 0.5$, but also $m/n = 0.4$ and $m/n = 0.6$. We need to emphasize the following point in our revised text. Relief is scale-invariant for $m/n = 0.5$, relief increases with scale for $m/n < 0.5$, but decreases with scale for $m/n > 0.5$. We can think of nothing about the morphodynamics of natural systems that would dictate such behavior.

**Reviewer 1**: *"The demonstrated scale invariance occurs in a model from which terms that impart scale dependence have been omitted. One such term is the diffusion term in equation 2 (which should be positive)."*

Thank you for pointing out the mistake in our equation; it has been fixed. We have purposely omitted the hillslope diffusion terms in order to study the behavior of SPIM itself. The use of hillslope diffusion resolves the problem of horizontal scale invariance for $m/n = 0.5$ only at the finest scales.

[Figure]

**Reviewer 1**: *"The authors argue that hillslope diffusion "operates only at small scales". Sure, but might that not contradict the conclusion that "the steady-state landscape for a 1 m² domain can be stretched so that it is identical to the corresponding landscape for a 100 km² domain?"*

No contradiction, just hard to get an acronym into the abstract. Here is a proposed rewriting. "Landscape evolution models often utilize the stream power incision model (SPIM) to simulate river incision. That is, the steady-state landscape predicted using SPIM alone for a 1 m² horizontal domain can be stretched so that it is identical to the corresponding landscape for a 100 km² domain."

**Reviewer 1**: *"The authors also do not consider channel width, another potential source of scale dependence."*

Width can indeed provide a source of scale dependence. We will point this out in a revised text. But the purpose of our paper is to study a 2D implementation of SPIM in the context of landscape evolution. SPIM does not predict channel width, and the addition of hillslope diffusion or hillslope length does not change this.

**Reviewer 1**: *"I appreciate what the authors are trying to do: discovering flaws in widely used models is one way that science advances. But they seem to construe their discovery as evidence that the entire community is asleep at the wheel, and I don't think that is true."*

We thank the reviewer for grasping the central point of our paper. The comment *"But they seem to construe their discovery as evidence that the entire community is asleep at the wheel, and I don't think that is true"* is more sociological than scientific. We

believe that our results stand on their own, and that the science speaks for itself without editorializing. It does not make sense, however, to point out the scale invariance issue when $m/n = 0.5$ without also pointing out that the use of this value is ubiquitous in the literature. (See table at the end of this response).

**Reviewer 1**: *"The fact that this simplification gives rise to scale invariance with a particular combination of parameters is indeed an odd quirk – one that is probably worthy of a cautionary tale – but it doesn't mean that the underlying arguments for relating incision rate to drainage area and slope are fundamentally flawed."*

We do not agree that our central result is an odd quirk. It is built into the fabric of SPIM. We repeat. Relief is scale-invariant for $m/n = 0.5$, relief increases with scale for $m/n < 0.5$, but decreases with scale for $m/n > 0.5$. We can think of nothing about the morphodynamics of natural systems that would dictate such behavior.

**Reviewer 1**: *"The version of the stream power model presented in this paper certainly has substantial limitations, and discussions of its shortcomings – as well as proposed improvements – abound in the literature."*

Yes, but... we have not found a single instance in the literature where the scale invariance associated with $m/n = 0.5$ has been recognized.

**Reviewer 1**: *"I see two ways in which the authors could potentially use their analysis to contribute to those discussions. First, perhaps they could show more clearly how scale-invariant models would lead researchers to draw incorrect conclusions about drainage basins, even if those researchers are aware of the limitations of the stream power model as a process law."*
Thank you for the suggestions. We agree that including these suggestions in our manuscript will greatly improve the impact and discussion of our manuscript. We hope to include examples of where the stream power incision model can lead to incorrect conclusions. Our first example of how SPIM can lead to incorrect conclusion is in its use to predict landscape relief. Whipple and Tucker [1999] show in a 1D model that SPIM can be used to predict landscape relief given the location of the channel head, $X_c$. In 2D models, the corresponding variable would be a critical area threshold, $A_c$, where above this threshold fluvial processes dominate (e.g. Montgomery and Dietrich 1988). We believe that our work on scaling relationships and ridge singularities can show that predicting relief using a 2D SPIM-based model cannot be reliable without a good understanding of what physical processes set the scale in landscapes (i.e. hillslope length/channel head location). Because the singularities affect the channel profile/relief most strongly in the headwaters, the fluvial relief of the landscape is sensitive to the choice of hillslope length. In determining the relief of the landscape, it could be that its value is more sensitive to the choice of hillslope length instead of the horizontal length of the basin (as predicted by SPIM). In addition, we believe our results also have implications for recent work on drainage basin reorganization and the stability of drainage divides. Most of the literature uses the $\chi$ methodology with SPIM to predict locations and stability of drainage divides. The stability of a drainage divide is taken to depend on the values of $\chi$ on either side of the drainage divide. $\chi$ is evaluated at a threshold value ($X_c$ or $A_c$) from the ridge, and like $\eta$, $\chi$ varies sensitively due to the singular behavior near the ridge. We believe that our analysis on the ridge singularities in both the 1D and 2D model can help elucidate the uncertainty in the prediction of stable drainage divide locations.

**Reviewer 1**: *"Second, they might consider whether the particular shortcomings they document offer any insights into how a better model of river incision could be*

*constructed – and whether any of the proposed improvements to the stream power law avoid these problems."*

In the literature, there have been many proposals for landscape evolution models that incorporate bedrock incision based on abrasion from saltating sediment particles. For example, Gasparini et al. 2007 propose a generalized saltaton-abrasion model. The steady state slope is given by the following equation $S = aA^{1-b}\left(1 - cA^{-0.5}\right)^{-1}$ (Note: We replace the actual parameters with simplified bulk parameters). If we non-dimensionalize this equation in the same manner as our manuscript, we get $\hat{S} = \frac{ag}{v^2}\hat{A}^{1-b}L^{3-2b}\left(1 - c\hat{A}^{-0.5}L^{-1}\right)^{-1}$. Just by inspection of this equation, we can see that it is impossible to make elevation invariant to horizontal length scale $L$. In addition, the exponent, $b$, formulation is made from empirical laws, where it is unlikely the term $3 - 2b \leq 0$. We will expand this discussion in a revised version of the manuscript to show that some bedrock incision models do not necessarily experience scale invariance.

We hope that additions such as the ones stated above will help our manuscript contribute to the discussion of the strengths and weakness of SPIM, and help motivate improved prediction of landscape evolution.

Gasparini, N. M., Whipple, K. X. and Bras, R. L.: Predictions of steady state and transient landscape morphology using sediment-flux-dependent river incision models, Journal of Geophysical Research, 112(F3), doi:10.1029/2006JF000567, 2007.

Montgomery, D. R. and Dietrich, W. E.: Where do channels begin?, Nature, 336(6196), 232–234, doi:10.1038/336232a0, 1988.

[Figure]

Whipple, K. X. and Tucker, G. E.: Dynamics of the stream-power river incision model: Implications for height limits of mountain ranges, landscape response timescales, and research needs, Journal of Geophysical Research: Solid Earth, 104(B8), 17661–17674, doi:10.1029/1999JB900120, 1999.

Please also note the supplement to this comment:
http://www.earth-surf-dynam-discuss.net/esurf-2017-15/esurf-2017-15-AC1-supplement.pdf

**ESurfD**

**Supplement:**

| Paper | Citation |
|---|---|
| Adams et al. 2017 | This paper details the LANDLAB v1.0 OverlandFlow component. "By default, $m_{sp}$ and $n_{sp}$ have set values of $m_{sp}$ = 0.5 and $n_{sp}$ = 1.0 that can be adjusted by the model user." |
| Braun and Willett, 2013 | Basis for the FastScape fluvial geomorphic model. The authors used $m/n$ = 0.5 for their sample solution. However, the authors do explore the effect of the value $n$ from 1.0 to 4.0 on the computational time needed to solve their implicit scheme. |
| Egholm et al., 2013 | $m/n$ = 0.5; however, there is unlikely to be scale invariance because their stream power incision model is more complex than the one we analyze. They employ a term that protects the bed from incision due to an alluvial cover. |
| Fox et al. 2014 | This paper presents an inversion method for backing out paleorock uplift rates in Taiwan. The analysis uses the ratio $m/n$ = 0.5. |
| Goren et al., 2014 | *Table 1* lists the default values where $m$ = 0.5 and $n$ = 1.0. Also uses $h$ = 2.0, which means $hm/n$ = 1.0. This paper is the basis for the DAC model. |
| Harel et al., 2016 | m/n = 0.51 +/- 0.14 from a global analysis. This value is not statistically significant from 0.5. |
| Hobley et al., 2017 | This paper details LANDLAB. "This is primarily to maintain dimensionally sensible units for K while still honoring the widely observed ratio of $m/n$ ~ 0.5, interpreted from channel concavities of natural rivers at apparent topographic steady state." |
| Passalacqua et al., 2006 | *Equation 2* shows a "special case of the general governing equation, widely used in landscape modeling [*e.g. Rodriguez-Iturbe and Rinaldo*, 1997] with $m/n \approx 0.5$." |
| Pelletier, 2004 | Uses the value $m/n$ = 0.5 to explore landscape evolution models with persistent drainage migration. |
| Willet et al., 2014 | In their **Response of χ to a Change in Drainage Area** section, they "assume that $h$ = 2 and $m/n$ = 0.5." The sample simulations using the DAC model uses $m$ = 0.5 and $n$ = 1.0. They also fit various values of $m/n$ when regressing chi vs. elevation plots in real drainage basins. |
| Whipple and Tucker, 1999 | "For typical values of the exponent…" in empirical relations they cite in their paper, "…the m/n ratio is predicted to fall into a narrow range near 0.5." This paper is widely cited when choosing an appropriate value of $m/n$. They state the range is between 0.3 and 0.6, near 0.5. |
| Whipple et al. 2017 | In this recent paper, the authors investigate whether low-relief, high-elevation surfaces are formed by preservation of relic landscapes or stream piracy, applied to the Tibetan Plateau. Their sample simulations are conducted with the stream power incision model with $m/n$ = 0.5 |
| Whipple et al. 2017 | This paper compares response timescales for divide migration and drainage capture. In all cases of their analysis, $m/n$ = 0.5. |
| Yang et al. 2015 | Their DAC 2D simulations used $n$ = 1 and $m$ = 0.5, but they used different values of $m/n$ for their χ profiles. They test a variety of $m/n$ values, but end up using m/n = 0.45. This paper also investigates the morphology of the Tibetan Plateau as does Whipple et al. 2017. |

Adams, J. M., Gasparini, N. M., Hobley, D. E. J., Tucker, G. E., Hutton, E. W. H., Nudurupati, S. S. and Istanbulluoglu, E.: The Landlab v1.0 OverlandFlow component: a Python tool for computing shallow-water flow across watersheds, Geoscientific Model Development, 10(4), 1645–1663, doi:10.5194/gmd-10-1645-2017, 2017.

Braun, J. and Willett, S. D.: A very efficient O(n), implicit and parallel method to solve the stream power equation governing fluvial incision and landscape evolution, Geomorphology, 180–181, 170–179, doi:10.1016/j.geomorph.2012.10.008, 2013.

Egholm, D. L., Knudsen, M. F. and Sandiford, M.: Lifespan of mountain ranges scaled by feedbacks between landsliding and erosion by rivers, Nature, 498(7455), 475–478, doi:10.1038/nature12218, 2013.

Fox, M., Goren, L., May, D. A. and Willett, S. D.: Inversion of fluvial channels for paleorock uplift rates in Taiwan, Journal of Geophysical Research: Earth Surface, 119(9), 1853–1875, doi:10.1002/2014JF003196, 2014.

Goren, L., Willett, S. D., Herman, F. and Braun, J.: Coupled numerical-analytical approach to landscape evolution modeling, Earth Surface Processes and Landforms, 39(4), 522–545, doi:10.1002/esp.3514, 2014a.

Harel, M.-A., Mudd, S. M. and Attal, M.: Global analysis of the stream power law parameters based on worldwide 10 Be denudation rates, Geomorphology, 268, 184–196, doi:10.1016/j.geomorph.2016.05.035, 2016.

Hobley, D. E. J., Adams, J. M., Nudurupati, S. S., Hutton, E. W. H., Gasparini, N. M., Istanbulluoglu, E. and Tucker, G. E.: Creative computing with Landlab: an open-source toolkit for building, coupling, and exploring two-dimensional numerical models of Earth-surface dynamics, Earth Surface Dynamics, 5(1), 21–46, doi:10.5194/esurf-5-21-2017, 2017.

Passalacqua, P., Porté-Agel, F., Foufoula-Georgiou, E. and Paola, C.: Application of dynamic subgrid-scale concepts from large-eddy simulation to modeling landscape evolution, Water Resources Research, 42(6), n/a-n/a, doi:10.1029/2006WR004879, 2006.

Pelletier, J. D.: Persistent drainage migration in a numerical landscape evolution model, Geophysical Research Letters, 31(20), doi:10.1029/2004GL020802, 2004.

Rodríguez-Iturbe, I. and Rinaldo, A.: Fractal River Basins: Chance and Self-Organization, Cambridge University Press, Cambridge., 1997.

Willett, S. D., McCoy, S. W., Perron, J. T., Goren, L. and Chen, C.-Y.: Dynamic Reorganization of River Basins, Science, 343(6175), 1248765–1248765, doi:10.1126/science.1248765, 2014.

Whipple, K. X. and Tucker, G. E.: Dynamics of the stream-power river incision model: Implications for height limits of mountain ranges, landscape response timescales, and research needs, Journal of Geophysical Research: Solid Earth, 104(B8), 17661–17674, doi:10.1029/1999JB900120, 1999.

Whipple, K. X., DiBiase, R. A., Ouimet, W. B. and Forte, A. M.: Preservation or piracy: Diagnosing low-relief, high-elevation surface formation mechanisms, Geology, 45(1), 91–94, doi:10.1130/G38490.1, 2017a.

Whipple, K. X., Forte, A. M., DiBiase, R. A., Gasparini, N. M. and Ouimet, W. B.: Timescales of landscape response to divide migration and drainage capture: Implications for the role of divide mobility in landscape evolution: Landscape Response to Divide Mobility, Journal of Geophysical Research: Earth Surface, 122(1), 248–273, doi:10.1002/2016JF003973, 2017b.

Yang, R., Willett, S. D. and Goren, L.: In situ low-relief landscape formation as a result of?river network disruption, Nature, 520(7548), 526–529, doi:10.1038/nature14354, 2015.

---

## Author Comment (AC2) · 23 Jun 2017

We thank the reviewer taking the time to review our paper. The comments will be very helpful to us for improving our manuscript.

**Reviewer 2**: *"This paper presents a call to arms, urging landscape evolution modelers who use the stream power incision model (SPIM) to "move on to more sophisticated models", which better represent the physical mechanisms responsible for river erosion of bedrock, such as abrasion by sediment."*

[Figure]

This is indeed a one of the main motivations for our manuscript. Thank you for recognizing the point.

**Reviewer 2**: *"The argument rests primarily on the finding of scale invariant solutions when the SPIM exponent ratio m/n = 0.5, for the case where the commonly-used hillslope "diffusion" term is omitted."*

While we believe that the scale invariant case is the most interesting and unexpected of our results, we think that our analysis of the slope and elevation singularities at the ridge and of the scaling when $m/n \neq 0.5$ are also important for our argument. In particular, relief increases with scale for $m/n < 0.5$, but decreases with scale for $m/n > 0.5$. We will make sure we emphasize its important in a revised version of the manuscript.

**Reviewer 2**: *"While I am sympathetic to the stated goals of this work, I worry that, ironically, this paper may have the opposite impact by focusing so narrowly on a rather anecdotal result."*

The horizontal scale invariance for $m/n = 0.5$ is indeed glaring. As documented immediately below, we suggest that this choice is not anecdotal, but instead reflects common usage in the landscape community. We need, however, to emphasize more clearly that our focus is not narrow, but covers the entire range of values of $m/n$. Repeating text above, relief is scale-invariant for $m/n = 0.5$, relief increases with scale for $m/n < 0.5$, but decreases with scale for $m/n > 0.5$. We can think of nothing about the morphodynamics of natural systems that would dictate such behavior.

**Reviewer 2**: *"The model behavior described here will rarely occur in model studies*

*because modelers typically use other m/n ratios, or hillslope diffusion terms, minimum hillslope lengths or other model components that avoid this result."*

We would like to suggest otherwise. Firstly, many modelers have indeed used the value m/n = 0.5, either as the sole value or as an option. Some notable examples are Willett et al. 2014 (use $m/n = 0.5$ and $hm/n = 0.5$), the FASTSCAPE MODEL (e.g. Braun and Willett 2013, use $m/n = 0.5$ and $hm/n = 0.5$), and LANDLAB (e.g. Hobley et al. 2017, use $m/n = 0.5$). While the values of both m and n can be altered in LANDLAB, $m/n = 0.5$ is set as the default. Our argument is when modelers have little information on what the $m/n$ ratio should be, their default value is 0.5, the value that leads to scale invariance. We provide a table of papers in which a value of $m/n$ equal or close to 0.5 has been used.

Secondly, neither the inclusion of a hillslope diffusion term nor the use of a minimum hillslope length rectifies the scale invariance problem associated with $m/n = 0.5$ in the larger sense. We refer back to middle three panels of Figure 2b of our manuscript. Shown therein are steady-state landscapes for $m/n = 0.5$, with horizontal scale $L_{2D} = 22.4$ km, 224 km and 2240 km. We assume for illustration that the "fine scale" length (diffusion or hillslope length) is 2 km. It follows that unrealistic scale invariance prevails over lengths corresponding to 91.1% of the smallest basin, 99.1% of the medium basin, and 99.9% of the largest basin. SPIM forces the landscape to behave like the bellows of an accordion; pushing scale down jacks up ALL the slopes when $m/n = 0.5$.

We emphasize our belief that it is important to study the behavior of SPIM itself without the use of other sub-models (e.g. hillslope diffusion). We further argue that insight into the fundamental behavior of SPIM will be valuable when choosing e.g. a bedrock abrasion-incision model for implementation within a landscape evolution model.
**Reviewer 2**: *"I agree with the suggestions of the first reviewer for how this work could be extended in constructive ways."*

The second reviewer also agrees with the first reviewer's suggestions for improving our work. As we said in the response to the first reviewer, we will add examples of conditions where SPIM leads to incorrect interpretations, and apply our scaling analysis to show that more sophisticated models do not suffer from horizontal scale invariance. Please look at our written response to the first review.

**Reviewer 2**: *"For example, can scale analysis be used to identify when the SPIM may lead to incorrect interpretations, or test the validity of divergent model outcomes, such as the findings of Egholm et al. (2013) who directly compared the SPIM with a bedload abrasion incision model?"*

Thank you for citing the Egholm et al. 2013 paper; this paper will be cited within our modified manuscript. This paper clearly features a problem that requires a model that is more sophisticated than SPIM. We appreciate the direct comparisons between SPIM and a bedload abrasion incision model. Egholm et al. [2013] uses many sub-models (e.g. hillslope diffusion, isostasy, landslides, etc.). Our paper, however, is focused on how SPIM captures incision in a 2D landscape model. Our paper will be improved by adding a comparison between the way in which relief structure created with a) SPIM and b) a bedrock-abrasion incision model scale with horizontal length. Thank you for your insightful comments, and we hope our proposed additions will satisfy your concerns regarding our paper.

Braun, J. and Willett, S. D.: A very efficient O(n), implicit and parallel method to solve the stream power equation governing fluvial incision and landscape evolution, Geomorphology, 180–181, 170–179, doi:10.1016/j.geomorph.2012.10.008, 2013.

Egholm, D. L., Knudsen, M. F. and Sandiford, M.: Lifespan of mountain ranges scaled by feedbacks between landsliding and erosion by rivers, Nature, 498(7455), 475–478, doi:10.1038/nature12218, 2013.

Hobley, D. E. J., Adams, J. M., Nudurupati, S. S., Hutton, E. W. H., Gasparini, N. M., Istanbulluoglu, E. and Tucker, G. E.: Creative computing with Landlab: an open-source toolkit for building, coupling, and exploring two-dimensional numerical models of Earth-surface dynamics, Earth Surface Dynamics, 5(1), 21–46, doi:10.5194/esurf-5-21-2017, 2017.

Willett, S. D., McCoy, S. W., Perron, J. T., Goren, L. and Chen, C.-Y.: Dynamic Reorganization of River Basins, Science, 343(6175), 1248765–1248765, doi:10.1126/science.1248765, 2014.

Whipple, K. X. and Tucker, G. E.: Dynamics of the stream-power river incision model: Implications for height limits of mountain ranges, landscape response timescales, and research needs, Journal of Geophysical Research: Solid Earth, 104(B8), 17661–17674, doi:10.1029/1999JB900120, 1999.

Please also note the supplement to this comment:
http://www.earth-surf-dynam-discuss.net/esurf-2017-15/esurf-2017-15-AC2-supplement.pdf

**Supplement:**

| Paper | Citation |
|---|---|
| Adams et al. 2017 | This paper details the LANDLAB v1.0 OverlandFlow component. "By default, $m_{sp}$ and $n_{sp}$ have set values of $m_{sp}$ = 0.5 and $n_{sp}$ = 1.0 that can be adjusted by the model user." |
| Braun and Willett, 2013 | Basis for the FastScape fluvial geomorphic model. The authors used $m/n$ = 0.5 for their sample solution. However, the authors do explore the effect of the value $n$ from 1.0 to 4.0 on the computational time needed to solve their implicit scheme. |
| Egholm et al., 2013 | $m/n$ = 0.5; however, there is unlikely to be scale invariance because their stream power incision model is more complex than the one we analyze. They employ a term that protects the bed from incision due to an alluvial cover. |
| Fox et al. 2014 | This paper presents an inversion method for backing out paleorock uplift rates in Taiwan. The analysis uses the ratio $m/n$ = 0.5. |
| Goren et al., 2014 | *Table 1* lists the default values where $m$ = 0.5 and $n$ = 1.0. Also uses $h$ = 2.0, which means $hm/n$ = 1.0. This paper is the basis for the DAC model. |
| Harel et al., 2016 | m/n = 0.51 +/- 0.14 from a global analysis. This value is not statistically significant from 0.5. |
| Hobley et al., 2017 | This paper details LANDLAB. "This is primarily to maintain dimensionally sensible units for K while still honoring the widely observed ratio of $m/n$ ~ 0.5, interpreted from channel concavities of natural rivers at apparent topographic steady state." |
| Passalacqua et al., 2006 | *Equation 2* shows a "special case of the general governing equation, widely used in landscape modeling [*e.g. Rodriguez-Iturbe and Rinaldo*, 1997] with $m/n \approx 0.5$." |
| Pelletier, 2004 | Uses the value $m/n$ = 0.5 to explore landscape evolution models with persistent drainage migration. |
| Willet et al., 2014 | In their **Response of χ to a Change in Drainage Area** section, they "assume that $h$ = 2 and $m/n$ = 0.5." The sample simulations using the DAC model uses $m$ = 0.5 and $n$ = 1.0. They also fit various values of $m/n$ when regressing chi vs. elevation plots in real drainage basins. |
| Whipple and Tucker, 1999 | "For typical values of the exponent…" in empirical relations they cite in their paper, "…the m/n ratio is predicted to fall into a narrow range near 0.5." This paper is widely cited when choosing an appropriate value of $m/n$. They state the range is between 0.3 and 0.6, near 0.5. |
| Whipple et al. 2017 | In this recent paper, the authors investigate whether low-relief, high-elevation surfaces are formed by preservation of relic landscapes or stream piracy, applied to the Tibetan Plateau. Their sample simulations are conducted with the stream power incision model with $m/n$ = 0.5 |
| Whipple et al. 2017 | This paper compares response timescales for divide migration and drainage capture. In all cases of their analysis, $m/n$ = 0.5. |
| Yang et al. 2015 | Their DAC 2D simulations used $n$ = 1 and $m$ = 0.5, but they used different values of $m/n$ for their χ profiles. They test a variety of $m/n$ values, but end up using m/n = 0.45. This paper also investigates the morphology of the Tibetan Plateau as does Whipple et al. 2017. |

Adams, J. M., Gasparini, N. M., Hobley, D. E. J., Tucker, G. E., Hutton, E. W. H., Nudurupati, S. S. and Istanbulluoglu, E.: The Landlab v1.0 OverlandFlow component: a Python tool for computing shallow-water flow across watersheds, Geoscientific Model Development, 10(4), 1645–1663, doi:10.5194/gmd-10-1645-2017, 2017.

Braun, J. and Willett, S. D.: A very efficient O(n), implicit and parallel method to solve the stream power equation governing fluvial incision and landscape evolution, Geomorphology, 180–181, 170–179, doi:10.1016/j.geomorph.2012.10.008, 2013.

Egholm, D. L., Knudsen, M. F. and Sandiford, M.: Lifespan of mountain ranges scaled by feedbacks between landsliding and erosion by rivers, Nature, 498(7455), 475–478, doi:10.1038/nature12218, 2013.

Fox, M., Goren, L., May, D. A. and Willett, S. D.: Inversion of fluvial channels for paleorock uplift rates in Taiwan, Journal of Geophysical Research: Earth Surface, 119(9), 1853–1875, doi:10.1002/2014JF003196, 2014.

Goren, L., Willett, S. D., Herman, F. and Braun, J.: Coupled numerical-analytical approach to landscape evolution modeling, Earth Surface Processes and Landforms, 39(4), 522–545, doi:10.1002/esp.3514, 2014a.

Harel, M.-A., Mudd, S. M. and Attal, M.: Global analysis of the stream power law parameters based on worldwide 10 Be denudation rates, Geomorphology, 268, 184–196, doi:10.1016/j.geomorph.2016.05.035, 2016.

Hobley, D. E. J., Adams, J. M., Nudurupati, S. S., Hutton, E. W. H., Gasparini, N. M., Istanbulluoglu, E. and Tucker, G. E.: Creative computing with Landlab: an open-source toolkit for building, coupling, and exploring two-dimensional numerical models of Earth-surface dynamics, Earth Surface Dynamics, 5(1), 21–46, doi:10.5194/esurf-5-21-2017, 2017.

Passalacqua, P., Porté-Agel, F., Foufoula-Georgiou, E. and Paola, C.: Application of dynamic subgrid-scale concepts from large-eddy simulation to modeling landscape evolution, Water Resources Research, 42(6), n/a-n/a, doi:10.1029/2006WR004879, 2006.

Pelletier, J. D.: Persistent drainage migration in a numerical landscape evolution model, Geophysical Research Letters, 31(20), doi:10.1029/2004GL020802, 2004.

Rodríguez-Iturbe, I. and Rinaldo, A.: Fractal River Basins: Chance and Self-Organization, Cambridge University Press, Cambridge., 1997.

Willett, S. D., McCoy, S. W., Perron, J. T., Goren, L. and Chen, C.-Y.: Dynamic Reorganization of River Basins, Science, 343(6175), 1248765–1248765, doi:10.1126/science.1248765, 2014.

Whipple, K. X. and Tucker, G. E.: Dynamics of the stream-power river incision model: Implications for height limits of mountain ranges, landscape response timescales, and research needs, Journal of Geophysical Research: Solid Earth, 104(B8), 17661–17674, doi:10.1029/1999JB900120, 1999.

Whipple, K. X., DiBiase, R. A., Ouimet, W. B. and Forte, A. M.: Preservation or piracy: Diagnosing low-relief, high-elevation surface formation mechanisms, Geology, 45(1), 91–94, doi:10.1130/G38490.1, 2017a.

Whipple, K. X., Forte, A. M., DiBiase, R. A., Gasparini, N. M. and Ouimet, W. B.: Timescales of landscape response to divide migration and drainage capture: Implications for the role of divide mobility in landscape evolution: Landscape Response to Divide Mobility, Journal of Geophysical Research: Earth Surface, 122(1), 248–273, doi:10.1002/2016JF003973, 2017b.

Yang, R., Willett, S. D. and Goren, L.: In situ low-relief landscape formation as a result of?river network disruption, Nature, 520(7548), 526–529, doi:10.1038/nature14354, 2015.

---

## Editor Comment (EC1) · J. Braun (Editor) · 31 Jul 2017

**Recommendation following revision**

(by Jean Braun, Associate Editor)

Manuscript entitled *Landscape evolution models using the stream power incision model show unrealistic behavior when $m/n$ equals 0.5*

by Kwang and Parker

I wish first to thank the authors for greatly improving their manuscript by taking into account most of the two reviewers suggestions. **Reviewer 1** argued that the peculiar behaviour of SPIM highlighted by Kwang and Parker is well known but is unlikely to be realized in nature. The authors adequately refute the first argument by noting that the scale invariance they highlight has not been adequately mentioned and studied in previous studies. They added a small section in their discussion to this effect. Concerning the second argument raised by the reviewer, they note that introducing a critical hillslope length scale does not resolve the issue of relief dependence on profile length and further renders the results of the SPIM strongly dependent on the choice of the critical length scale. They added an additional section (8) in the manuscript describing this in details. **Reviewer 1** also stated that the author should widen their study to describe whether other landscape evolution models suffer from the same singluarity and/or scaling behaviour. The authors responded by adding a new section (7) to their manuscript where they show that another lesser utilized model by Gasparini et al (2007) does not suffer from the problems concerning the SPIM. The authors responded adequately to the Reviewer's comments concerning the ommision of hillslope diffusion and channel width dynamics in their model. **Reviewer 1** also remarked that the behaviour described by the authors is limited to a peculiar choice of the ratio $m/n$, but the authors responded that the relationship between scale and relief is odd for most values of the ratio $m/n$. I do follow the authors on this point too, although I woud have appreciated that they further document their statement that *We can think of nothing about the morphodynamics of natural systems that would dictate such a behaviour*. Do we know what the natural system behaviour is? Does relief increase or decrease with scale, everything else being kept constant? A short section/sentence on how relief scale with the horizontal scale of the system in areas dominated by bedrock incision would be welcome. Finally **Reviewer 1** made two suggestions. The first concerns the consequences of using a scale-invariant model and the second whether a better model could be designed that does not suffer from scale invariance. The authors responded positively to both suggestions by adding a long section in the discussion concerning the first point and a new section concerning the second. In both cases, their arguments are valid and, in my opinion, greatly improve the impact of the manuscript.

**Reviewer 2** also questions the importance of the authors' findings by calling it an *anecdotal result*. The authors argue that the value of $m/n = 0.5$ is the most commonly used value in the literature. To make the point they added a list (as supplementary material) of key papers where $m/n = 0.5$ has been used. **Reviewer 2** argued that the inclusion of a critical hillslope length scale or diffusion term is commonly used

to remedy the problem highlighted by the authors. As discussed above the authors refuted this point by inserting an adequate discussion in the manuscript on the effect of adding diffusion or a critical length scale. As proposed by Reviewer 1, **Reviewer 2** also suggested that the authors find ways to render their manuscript more *useful* and *positive* by highlighting the consequences of using $m/n$ and proposing alternative models. The authors have, in my opinion, responded positively to these suggestions (see response to Reviewer 1 above).

In conclusion, I believe that the authors have very positively and adequately responded to the two reviers comments, critics and suggestions. It is comforting to see that both reviewers made very similar remarks, which, in my opinion, really helped to improve the manuscript.

It is also my opinion that this manuscript highlights an interesting behaviour of the most commonly used equation for large-scale landscape evolution (at least in situations where bedrock incision dominates) that deserves to be published. Despite the fact that this behaviour is (more or less) known to those actively working with SPIM and its implementation in numerical models, it deserves to be made clearer to the community. This manuscript can also be seen now as a warning to all potential users of the potential *unnatural* consequences of using SPIM. In its improved/modified form, the manuscript also points to the existence of other representations of bedrock incision that do not suffer from this scale invariance and singular behaviour. It should, therefore, in my opinion, be sent to **Reviewer 2** who has asked to see the revised version and, unless he/she indicates that further modifications are needed to improve the impact of the manuscript, it should be accepted pending minor revisions. These revisions should include a short discussion on what we know about the scaling relationship between system length and relief in natural systems (see my comment above).

---

## Author Response (AR2)

October 17, 2017

Dear Editors,

We made minor revisions to our revised manuscript. The main revisions are in section 9, *Discussion and conclusion*.

1. The associate editor requested that we make revisions that included "a short discussion on what we know about the scaling relationship between system length and relief in natural systems." We added a brief discussion in the *Discussion and conclusion* section of our revised manuscript.

2. In addition, we included a citation (Gasparini and Brandon, 2011) based on a discussion that we had about the distinction between the field measurements of the $m/n$ ratio and those used in SPIM.

We hope these revisions will help our manuscript meet the requirements for publication in ESurf.

Best,

Jeffrey Kwang and Gary Parker

[revised manuscript text omitted]